# Genome taxonomy of the genus *Thalassotalea* and proposal of *Thalassotalea hakodatensis* sp.nov. isolated from sea cucumber larvae

Ryota Yamano[1], Juanwen Yu[1], Alfabetian Harjuno Condro Haditomo[1,2], Chunqi Jiang[1,3], Sayaka Mino[1], Jesús L. Romalde[4], Kyuhee Kang[5], Yuichi Sakai[6], Tomoo Sawabe[1]*

1 Laboratory of Microbiology, Faculty of Fisheries Sciences, Hokkaido University, Hakodate, Japan, 2 Aquaculture Department, Faculty of Fisheries and Marine Sciences, Universitas Diponegoro, Semarang, Indonesia, 3 Atmosphere and Ocean Research Institute, University of Tokyo, Chiba, Japan, 4 Departamento de Microbiología y Parasitología, CRETUS & CIBUS-Facultad de Biología, Universidade de Santiago de Compostela, Santiago, Spain, 5 Korean Collection for Type Cultures (KCTC), Biological Resource Center, Korea Research Institute of Bioscience and Biotechnology, Jeongeup-si, Jeollabuk-do, South Korea, 6 Hakodate Fisheries Research, Hokkaido Research Organization, Local Independent Administrative Agency, Hakodate, Japan

* sawabe@fish.hokudai.ac.jp

**Data Availability Statement:** The GenBank accession number for the 16S rRNA gene sequence of the type strain PTE2 is LC757706. The whole genome sequence of the PTE2 strain and

## Abstract

The genus *Thalassotalea* is ubiquitous in marine environments, and up to 20 species have been described so far. A Gram-staining-negative, aerobic bacterium, designated strain PTE2[T] was isolated from laboratory-reared larvae of the Japanese sea cucumber *Apostichopus japonicus*. Phylogenetic analysis based on the 16S rRNA gene nucleotide sequences revealed that PTE2[T] was closely related to *Thalassotalea sediminis* N211[T] (= KCTC 42588[T] = MCCC 1H00116[T]) with 97.9% sequence similarity. ANI and *in silico* DDH values against *Thalassotalea* species were 68.5–77.0% and 19.7–24.6%, respectively, indicating the novelty of PTE2[T]. Based on genome-based taxonomic approaches, strain PTE2[T] (= JCM 34608[T] = KCTC 82592[T]) is proposed as a new species, *Thalassotalea hakodatensis* sp. nov.

## Introduction

The genus *Thalassotalea*, a member of the family *Colwelliaceae* in the order *Alteromonadales*, was first proposed by Zhang et al. [1] with the description of *Thalassotalea piscium*. At the same time, four *Thalassomonas* species, *Thalassomonas ganghwensis* [2], *Thalassomonas loyana* [3], *Thalassomonas agarivorans* [4] and *Thalassomonas agariperforans* [5], were reclassified into the genus *Thalassotalea* as *Thalassotalea ganghwensis*, *Thalassotalea loyana*, *Thalassotalea agarivorans* and *Thalassotalea agariperforans*, respectively. Currently, 20 species have been described in this genus [1, 6–18]. *Thalassotalea* strains have been isolated from flounder [1], seawater [4, 6, 15, 19], marine sediment [2, 8, 16, 20], marine sand [5, 12], corals [3, 9, 10, 14], aquaculture systems [7], pacific oyster [11], deep-sea seamounts [13], mangrove sediment [17] and red alga [18]. The genus is characterized as being rod-shaped, Gram-negative, aerobic

seven Thalassotalea type species shown in Table 1 have been deposited to DDBJ/ENA/GenBank under the accession numbers AP027361-AP027365, BSST01000001-BSST01000003, BSSU01000001-BSSU01000035, and BSSV01000001-BSSV01000017. Raw reads for genome assembly used in this study have been deposited under DRA015858.

**Funding:** This study was supported by Kaken 19K22262. The funders had no role in study design, data collection and analysis, decision to publish, or preparation of the manuscript.

**Competing interests:** The authors have declared that no competing interests exist.

or facultatively anaerobic, catalase and oxidase-positive, and motile with single polar flagellum or non-motile. A previous study showed that one of the members of this genus, *Thalassotalea* sp. PP2-459 isolated from a carpet-shell clam, possesses quorum-quenching ability against *Vibrio anguillarum*, which is one of the key pathogens in marine bivalve aquaculture, and thus this strain is proposed as a probiotic candidate [21]. The strain is also known to produce *N*-acyl dehydrotyrosines, which shows the potential for therapeutic and cosmetic applications [22]. In addition, members of *Thalassotalea* may contribute to nutrient cycling in marine environments by their polysaccharide degrading abilities [23]. Nevertheless, despite the ubiquity of the *Thalassotalea* strains and their heterogeneity in phenotypes, no comprehensive genome studies have been performed yet.

During studies on microbiomes associated with the early life stages of the sea cucumber *Apostichopus japonicus* [24], a new species candidate, strain PTE2, of the genus *Thalassotalea* was isolated from the pentactula larvae. The authors report that ASVs (ASV0402, ASV0238, and ASV0323), which possess the same V1-V2 region as strain PTE2, increased in abundance in the seawater of late auricularia and pentacutula larvae, suggesting that strain PTE2 may play an important role in host-microbe interaction during sea cucumber development [24]. In this study, modern polyphasic taxonomic studies was employed, including molecular phylogenetic analysis based on 16S rRNA gene nucleotide sequences, phenotypic characterization and genome comparisons, to characterize the newly described species, *Thalassotalea* sp. PTE2$^T$ (= JCM 34608$^T$ = KCTC 82592$^T$).

## Materials and methods

### Bacterial strains and phenotypic characterization

The strain PTE2$^T$ was isolated from the pentactula larvae of *Apostichopus japonicus* [24]. Bacterial colonies were purified using a 1/5 strength ZoBell 2216 agar or broth [24]. *T. sediminis* KCTC 42588$^T$, *T. insulae* KCTC 62186$^T$, *T. piscium* JCM 18590$^T$, *T. agarivorans* JCM 13379$^T$, and *T. atypica* JCM 31894$^T$ were used as references for genomic and phenotypic comparisons against the strain PTE2$^T$. All strains were cultured on Marine agar 2216 (BD, Franklin Lakes, New Jersey, USA). The phenotypic characteristics were determined according to previously described methods [25–29]. Motility was observed under a microscope using cells suspended in droplets of sterilized 75% artificial seawater (ASW).

### Molecular phylogenetic analysis based on 16S rRNA gene nucleotide sequences

The almost full length 16S rRNA gene sequence (1,424 bp) of strain PTE2$^T$ was obtained by direct sequencing of PCR-amplified DNA. 27F and 1509R were used as amplification primers, and four primers: 27F, 800F, 920R and 1509R were used for the Sanger sequencing [29, 30]. Raw sequence reads were assembled to generate a contig using ChromasPro Ver.2.1.10 (Technelysium Pty. Ltd. South Brisbane, Australia). The 16S rRNA gene nucleotide sequences of the type strains of the genus *Thalassotalea* and other *Colwelliaceae* species were retrieved from NCBI databases. Sequences were aligned using Silva Incremental Aligner v1.2.11 [29, 31]. A phylogenetic model test and maximum likelihood (ML) tree reconstruction were performed using the MEGAX v.10.1.8 program [29, 32, 33]. ML tree was reconstructed with 1,000 bootstrap replications using Kimura 2-parameter (K2) with gamma distribution (+G) and invariant site (+I) model. In addition, nucleotide similarities among strains were also calculated using "compute pairwise distance" in MEGAX.

## Whole genome sequencing

Genomic sequence of PTE2[T], *T. sediminis* KCTC 42588[T], *T. agarivorans* JCM 13379[T], *T. piscium* JCM 18590[T] *T. insulae* KCTC 62186[T] and *T. atypica* JCM 31894[T] was obtained using a hybrid assembly method described previously [29]. Genomic sequence of *T. loyana* LMG 22536[T] and *T. eurytherma* JCM 18482[T] was obtained by using only Illumina sequence reads. Illumina sequence was obtained using a method described previously [29], and assembled using Unicycler 0.4.8 [34]. The whole genome sequences were annotated with DDBJ Fast Annotation and Submission Tool (DFAST) [35]. The complete genome sequences of PTE2[T] acquired in this study were deposited in GenBank/EMBL/DDBJ under accession number AP027365.

## Overall genome relatedness indices (OGRIs)

Overall genome relatedness indices (OGRIs) were calculated to determine the novelty of PTE2[T] using same methodology described in Yamano et al. [29]. Average nucleotide identities (ANIs) were calculated using the Orthologous Average Nucleotide Identity Tool (OrthoANI) software to calculate OrthoANI with a default setting [36] using genomes of the PTE2[T] [29]. *In silico* DDH values were calculated using Genome-to-Genome Distance Calculator (GGDC) 2.1 based on formula 2 being the most robust formula using both complete and incomplete genomes [37]. Average amino acid identities (AAIs) were calculated between PTE2[T] and other related *Colwelliaceae* species (Tables 1 and S1) using an enveomics toolbox [38].

## Multilocus sequence analysis (MLSA)

MLSA was performed as previously described [25, 26, 29, 39, 40]. The sequences of four protein-coding genes (*ftsZ*, *mreB*, *rpoA*, and *topA*), essential single-copy genes in the taxa examined in this study were obtained from the genome sequences of PTE2[T], *T. sediminis* KCTC 42588[T], *T. insulae* KCTC 62186[T], *T. piscium* JCM 18590[T], *T. agarivorans* JCM 13379[T], *T. loyana* LMG 23556[T], *T. eurytherma* JCM 18482[T], *T. atypica* JCM 31894[T], *T. marina* QBLM2[T], *T. profundi* YM155[T], *T. mangrovi* zs-4[T], *T. crassostreae* LPB0090[T], *T. algicola* M1351[T], *T. litorea* MCCC IK03283, *T. euphylliae* H2 and other related *Colwelliaceae* and *Idiomarinaceae* species (Tables 1 and S1) (see genome accession number in the description section below). The sequences of each gene were aligned using ClustalX 2.1 [41]. Concatenation of sequences and phylogenetic NeighborNet reconstruction were performed using SplitsTree 4.16.2 with options of JukesCantor correction, gap-exclusion and 1,000 bootstrap [42]. Regions used for the network reconstruction in Fig 3 were 1–2,549, 1–1,044 1–2,019, and 1–2,549 for *ftsZ*, *mreB*, *rpoA*, and *topA*, as PTE2 nucleotide sequence positions, respectively.

## Pan and core genome analysis

A total of 15 genomes, including eight newly obtained genomes in this study (PTE2[T], *T. sediminis* JCM 42588[T], *T. insulae* KCTC 62186[T], *T. piscium* JCM 18590[T], *T. agarivorans* JCM 13379[T], *T. loyana* LMG 23556[T], *T. eurytherma* JCM 18482[T], *T. atypica* JCM 31894[T]) and seven retrieved from the NCBI database (*T. marina* QBLM2[T], *T. profundi* YM155[T], *T. mangrovi* zs-4[T], *T. crassostreae* LPB0090[T], *T. algicola* M1351[T], *T. litorea* MCCC IK03283, *T. euphylliae* H2), were used for pangenome analysis using the program anvi'o v7 [43] based on previous studies [28, 29, 44–46], with minor modifications. Briefly, contig databases of each genome were constructed by fasta files (anvi-gen-contigs-database) and decorated with hits from HMM models (anvi-run-hmms). Subsequently, functions were annotated for genes in contig databases (anvi-run-ncbi-cogs). KEGG annotation was also performed (anvi-run-kegg-

kofams) [29]. The storage database was generated (anvi-gen-genomes-storage) using all contigs databases and pangenome analysis was performed (anvi-pan-genome) [29]. The results were displayed (anvi-display-pan) and adjusted manually [29].

### *In silico* chemical taxonomy: Prediction of fatty acids, polar lipids and isoprenoid quinone using the comparative genomics approach

The genes encoding key enzymes and proteins for the synthesis of fatty acids (FAs), polar lipids and isoprenoid quinones were retrieved from the genome sequences of PTE2[T] and the related species using *in silico* MolecularCloning ver. 7 with same methodology by Yamano et al. [29]. Genomic structure and distribution of the genes were compared also using *in silico* MolecularCloning ver. 7. The 3D structures of genes encoding FA desaturase from some of the strains were predicted using Phyre2 [47].

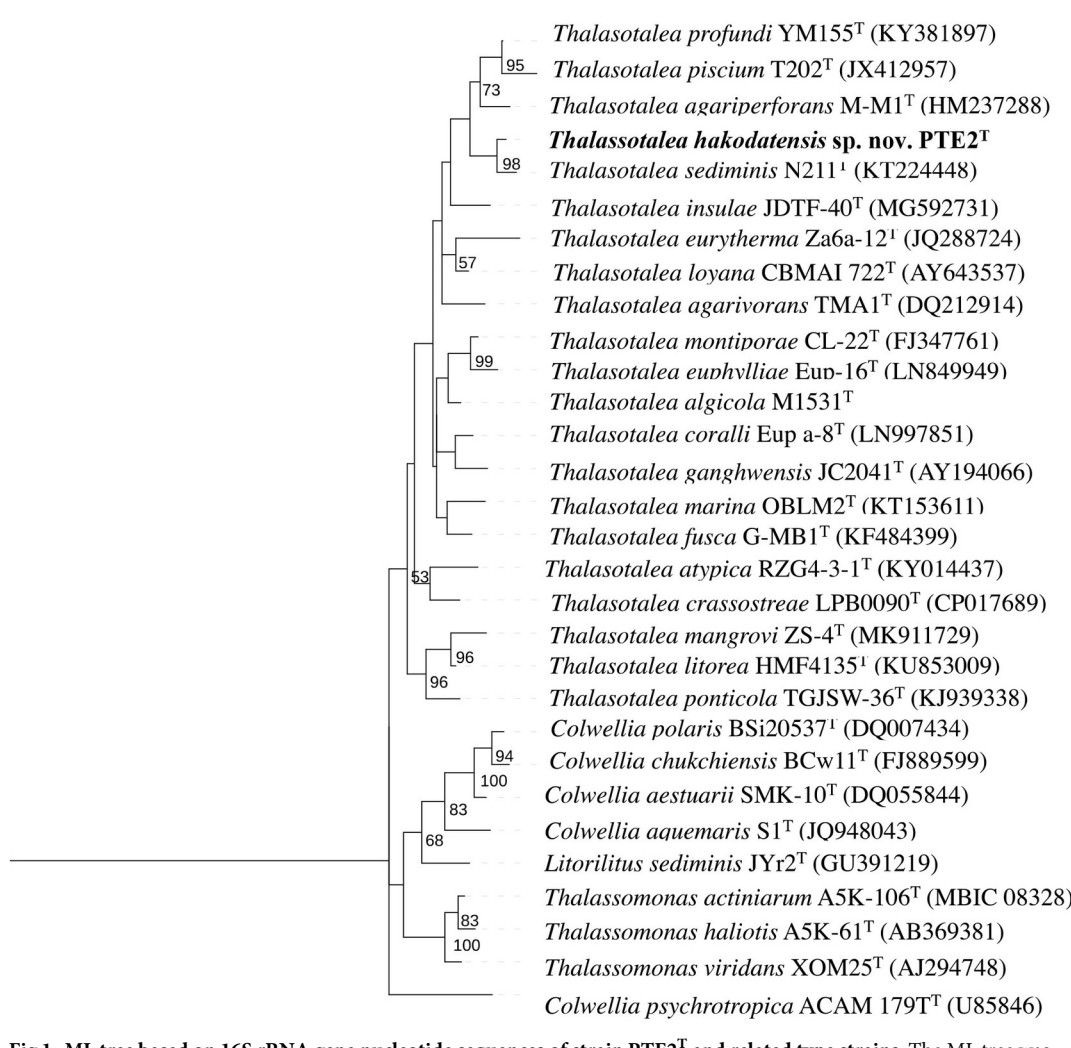

**Fig 1. ML tree based on 16S rRNA gene nucleotide sequences of strain PTE2[T] and related type strains.** The ML tree was reconstructed using the K2+G+I model. Numbers shown on branches are bootstrap values (%) based on 1,000 replicates (>50%). 1,347 bp were compared (69–1,416 position in *T. sediminis* N2111[T], KT224448). *Idiomarina ramblicola* R22[T], of which position was not visible in Fig 1, was used as an outgroup to generate this rooted tree.

# Results and discussion

## Molecular phylogenetic analysis based on 16S rRNA gene nucleotide sequences

Phylogenetic analysis based on 16S rRNA gene nucleotide sequences showed that strain PTE2[T] could be affiliated to the members of the genus *Thalassotalea* (Fig 1). However, the internal node of the genus *Thalassotalea* was unlikely to be supported by a high bootstrap value (see also MLSA section). The strain showed the highest sequence similarities of 97.9% with *T. sediminis*, which is below the proposed threshold range of the species boundary, 98.7% [48, 49].

## Genomic features and overall genome relatedness indices (OGRIs)

Genomic features of PTE2[T] and the described *Thalassotalea* species with available genomic sequences were shown in (Table 1). The complete genomic sequence of PTE2[T] showed that genome size and G+C content is 4.31 Mb and 38.5%, respectively. *In silico* DDH and ANI values of the PTE2[T] against 14 *Thalassotalea* species were 9.7–24.6% and 68.5–77.0% respectively, which were below the species boundary threshold of 70% and 95% proposed in previous studies (S2 Table) [29]. AAI values were between 59.8–80.5% (Fig 2), which were also below the species delineation boundary of 95–96% [50], confirming that strain PTE2[T] represents a novel species in the genus *Thalassotalea*. Interestingly, while five *Thalassotalea* species (*T. sediminis*, *T. marina*, *T. insulae*, *T. piscium* and *T. profundi*) showed relatively high AAI values to strain PTE2[T], other *Thalassotalea* species showed lower AAI values than species in *Colwellia*, *Cognaticolwellia*, *Pseudocolwellia*, *Litorilituus* or *Thalassomonas*.

## Multilocus sequence analysis (MLSA)

MLSA network showed that PTE2[T] is a novel species different from the 14 reference *Thalassotalea* strains used in this study. This analysis also showed that PTE2[T], *T. insulae*, *T. marina T. piscium*, *T. profundi*, and *T. sediminis*, form a monophyletic clade of the genus *Thalassotalea* (Fig 3). The MLSA network analysis revealed intriguing findings about the *Thalassotalea* species and the *Colwellia* genus. Specifically, the other *Thalassotalea* species did not form a

**Table 1. Genome properties of PTE2[T] and *Thalassotalea* species with available genomic sequences.**

| Species | Strain | RefSeq accession | Size | Genome assemble status | GC content |
|---|---|---|---|---|---|
| *Thalassotalea hakodatensis* sp. nov. | PTE2[T] | AP027365 (in this study) | 4.31 Mb | complete (1 chromosome) | 38.5% |
| *Thalassotalea sediminis* | KCTC 42588[T] | AP027361 (in this study) | 3.90 Mb | complete (1 chromosome) | 38.6% |
| *Thalassotalea insulae* | KCTC 62186[T] | BSST01000001-BSST01000003 (in this study) | 4.39 Mb | draft (3 contigs) | 40.3% |
| *Thalassotalea piscium* | JCM 18590[T] | AP027362 (in this study) | 3.90 Mb | complete (1 chromosome) | 37.5% |
| *Thalassotalea agarivorans* | JCM 13379[T] | AP027363 (in this study) | 3.39 Mb | complete (1 chromosome) | 41.9% |
| *Thalassotalea loyana* | LMG 22536[T] | BSSV01000001-BSSV01000017 (in this study) | 4.00 Mb | draft (17 contigs) | 40.4% |
| *Thalassotalea eurytherma* | JCM 18482[T] | BSSU01000001-BSSU01000035 (in this study) | 3.55 Mb | draft (35 contigs) | 39.7% |
| *Thalassotalea atypica* | JCM 31894[T] | AP027364 (in this study) | 4.41 Mb | complete (1 chromosome) | 40.2% |
| *Thalassotalea marina* | QBLM2[T] | GCF 014656435.1 | 4.87 Mb | draft (31 contigs) | 39.4% |
| *Thalassotalea profundi* | YM155[T] | GCF 014653195.1 | 3.99 Mb | draft (40 contigs) | 36.3% |
| *Thalassotalea mangrovi* | zs-4[T] | GCF 005116735.1 | 3.71 Mb | draft (76 contigs) | 45.9% |
| *Thalassotalea crassostreae* | LPB0090[T] | GCF 001831495.1 | 3.86 Mb | complete (1 chromosome) | 38.8% |
| *Thalassotalea algicola* | M1531[T] | GCF_012932965.1 | 4.06 Mb | draft (19 contigs) | 39.1% |
| *Thalassotalea litorea* | MCCC IK03283 | GCF 005116735.1 | 3.88 Mb | draft (46 contigs) | 43.9% |
| *Thalassotalea euphylliae* | H2 | GCF 003390395.1 | 4.36 Mb | complete (1 chromosome) | 43.0% |

Fig 2. AAI matrix using *Thalassotalea* and *Colwelliaceae* genomes.

| | T. hakodatensis PTE2$^T$ | T. sediminis | T. marina | T. insulae | T. piscium | T. profundi | Col. chukchiensis | Cog. beringensis | Cog. aestuarii | Col. polaris | Col. hornerae | T. algicola | Cog. mytili | T. euphylliae | P. agarivorans | L. sediminis | T. eurytherma | T. agarivorans | T. atypica | L. lipolyticus | T. loyana | Col. ponticola | Tm. actinarium | Col. marinimaniae | Tm. viridans | Col. echini | Col. psycherythraea | Col. demingiae | Col. piezophila | T. litorea | T. mangrovi | T. crassostreae |
|---|---|---|---|---|---|---|---|---|---|---|---|---|---|---|---|---|---|---|---|---|---|---|---|---|---|---|---|---|---|---|---|---|
| *Thalassotalea hakodatensis* sp. nov. PTE2$^T$ | 100 | 80.5 | 70.0 | 68.9 | 66.9 | 66.8 | 65.7 | 64.9 | 64.9 | 64.6 | 64.6 | 64.4 | 64.3 | 63.6 | 63.5 | 63.3 | 63.3 | 63.2 | 63.1 | 63.1 | 62.8 | 62.7 | 62.4 | 62.4 | 62.3 | 61.7 | 61.3 | 61.3 | 61.2 | 60.1 | 59.8 | 59.8 |
| *Thalassotalea sediminis* | 80.5 | 100 | 70.5 | 69.0 | 66.6 | 66.8 | 64.9 | 64.7 | 64.9 | 64.9 | 64.7 | 64.1 | 64.7 | 63.4 | 63.5 | 63.6 | 63.4 | 63.2 | 63.6 | 63.3 | 63.0 | 62.8 | 62.8 | 62.1 | 62.7 | 61.7 | 61.8 | 62.0 | 61.5 | 60.2 | 60.2 | 60.4 |
| *Thalassotalea marina* | 70.0 | 70.5 | 100 | 67.7 | 66.0 | 65.8 | 65.2 | 63.9 | 63.8 | 63.9 | 63.8 | 63.1 | 63.5 | 63.0 | 62.4 | 62.8 | 63.1 | 62.7 | 63.0 | 62.7 | 62.4 | 62.5 | 61.5 | 61.7 | 61.4 | 60.6 | 61.0 | 61.1 | 60.8 | 59.7 | 59.7 | 59.9 |
| *Thalassotalea insulae* | 68.9 | 69.0 | 67.7 | 100 | 68.3 | 68.3 | 66.8 | 66.7 | 66.6 | 66.3 | 66.6 | 65.0 | 66.5 | 64.1 | 64.5 | 64.9 | 64.2 | 63.7 | 64.8 | 64.2 | 63.4 | 64.4 | 64.2 | 63.9 | 64.1 | 62.8 | 63.3 | 63.4 | 63.0 | 60.6 | 60.8 | 61.0 |
| *Thalassotalea piscium* | 66.9 | 66.6 | 66.0 | 68.3 | 100 | 80.4 | 67.7 | 66.9 | 67.2 | 66.7 | 66.6 | 64.8 | 66.5 | 64.3 | 65.5 | 64.4 | 63.9 | 63.1 | 64.4 | 64.4 | 63.5 | 64.7 | 64.3 | 63.6 | 63.8 | 63.9 | 63.6 | 63.9 | 63.6 | 60.4 | 60.2 | 60.6 |
| *Thalassotalea profundi* | 66.8 | 66.8 | 65.8 | 68.3 | 80.4 | 100 | 67.0 | 66.5 | 66.8 | 66.5 | 66.8 | 64.2 | 66.2 | 63.5 | 65.6 | 64.5 | 63.6 | 63.2 | 64.6 | 64.6 | 63.1 | 64.6 | 63.7 | 63.8 | 63.2 | 63.5 | 63.6 | 63.4 | 63.1 | 60.2 | 60.1 | 60.2 |
| *Colwellia chukchiensis* | 65.7 | 64.9 | 65.2 | 66.8 | 67.7 | 67.0 | 100 | 77.6 | 78.2 | 76.6 | 67.3 | 64.7 | 75.5 | 64.0 | 64.9 | 65.2 | 64.2 | 63.1 | 64.8 | 65.2 | 63.8 | 64.9 | 64.7 | 64.6 | 64.2 | 63.2 | 64.4 | 64.2 | 64.2 | 60.2 | 60.3 | 60.8 |
| *Cognaticolwellia beringensis* | 64.9 | 64.7 | 63.9 | 66.7 | 66.9 | 66.5 | 77.6 | 100 | 80.5 | 83.9 | 69.6 | 64.1 | 77.9 | 63.2 | 65.8 | 64.6 | 63.5 | 62.3 | 64.3 | 64.6 | 63.5 | 66.9 | 64.5 | 64.7 | 63.8 | 64.3 | 65.2 | 65.1 | 64.6 | 60.0 | 59.8 | 60.6 |
| *Cognaticolwellia aestuarii* | 64.9 | 64.9 | 63.8 | 66.6 | 67.2 | 66.8 | 78.2 | 80.5 | 100 | 79.7 | 68.7 | 64.0 | 79.6 | 63.2 | 66.8 | 65.5 | 63.4 | 62.8 | 64.8 | 65.0 | 62.9 | 66.1 | 64.9 | 64.7 | 64.6 | 64.2 | 65.0 | 65.5 | 64.2 | 59.7 | 59.4 | 61.0 |
| *Colwellia polaris* | 64.6 | 64.9 | 63.9 | 66.3 | 66.7 | 66.5 | 76.6 | 83.9 | 79.7 | 100 | 69.2 | 64.1 | 76.9 | 63.4 | 66.4 | 64.9 | 63.4 | 62.2 | 64.3 | 64.5 | 63.1 | 66.0 | 64.2 | 64.9 | 63.7 | 64.5 | 64.6 | 64.5 | 64.1 | 59.7 | 59.7 | 60.6 |
| *Colwellia hornerae* | 64.6 | 64.7 | 63.8 | 66.6 | 66.8 | 66.8 | 67.3 | 69.6 | 68.7 | 69.2 | 100 | 63.6 | 68.5 | 63.0 | 66.1 | 65.3 | 63.3 | 62.2 | 64.5 | 65.4 | 62.5 | 67.1 | 64.4 | 65.6 | 64.0 | 64.9 | 66.7 | 66.6 | 65.4 | 59.4 | 59.3 | 60.3 |
| *Thalassotalea algicola* | 64.4 | 64.1 | 63.1 | 65.0 | 64.8 | 64.2 | 64.7 | 64.1 | 64.0 | 64.1 | 63.6 | 100 | 63.9 | 73.6 | 62.9 | 64.2 | 65.8 | 63.8 | 68.5 | 64.0 | 65.1 | 62.7 | 63.6 | 62.8 | 63.3 | 61.8 | 62.2 | 62.4 | 62.1 | 60.0 | 60.7 | 61.0 |
| *Cognaticolwellia mytili* | 64.3 | 64.7 | 63.5 | 66.5 | 66.5 | 66.2 | 75.5 | 77.9 | 79.6 | 76.9 | 68.5 | 63.9 | 100 | 62.9 | 65.8 | 64.7 | 63.6 | 62.7 | 64.6 | 65.0 | 62.7 | 65.4 | 64.5 | 65.2 | 64.3 | 63.4 | 65.3 | 65.5 | 64.0 | 59.6 | 59.6 | 60.8 |
| *Thalassotalea euphylliae* | 63.6 | 63.4 | 63.0 | 64.1 | 64.3 | 63.5 | 64.0 | 63.2 | 63.2 | 63.4 | 63.0 | 73.6 | 62.9 | 100 | 62.2 | 63.4 | 65.0 | 63.5 | 67.0 | 62.9 | 64.3 | 62.8 | 62.5 | 62.0 | 62.4 | 61.3 | 61.5 | 61.7 | 61.2 | 60.6 | 60.5 | 60.4 |
| *Pseudocolwellia agarivorans* | 63.5 | 63.5 | 62.4 | 64.5 | 65.5 | 65.6 | 64.9 | 65.8 | 66.8 | 66.4 | 66.1 | 62.9 | 65.8 | 62.2 | 100 | 63.1 | 62.4 | 62.1 | 62.9 | 63.4 | 62.1 | 64.1 | 63.7 | 62.5 | 63.6 | 63.2 | 63.2 | 62.6 | 58.7 | 58.8 | 59.7 |
| *Litorilituus sediminis* | 63.3 | 63.6 | 62.8 | 64.9 | 64.4 | 64.5 | 65.2 | 64.6 | 65.5 | 64.9 | 65.2 | 64.2 | 64.7 | 63.4 | 63.1 | 100 | 63.3 | 62.6 | 64.4 | 75.6 | 62.5 | 70.7 | 64.0 | 70.6 | 63.9 | 67.8 | 70.5 | 70.5 | 70.1 | 59.5 | 59.7 | 60.4 |
| *Thalassotalea eurytherma* | 63.3 | 63.4 | 63.1 | 64.2 | 63.9 | 63.6 | 64.2 | 63.5 | 63.4 | 63.4 | 63.3 | 65.8 | 63.6 | 65.0 | 62.4 | 63.3 | 100 | 63.5 | 64.9 | 63.2 | 79.9 | 62.6 | 63.1 | 62.3 | 62.4 | 61.5 | 61.7 | 61.9 | 61.6 | 60.6 | 60.4 | 60.9 |
| *Thalassotalea agarivorans* | 63.2 | 63.2 | 62.7 | 63.7 | 63.1 | 63.2 | 63.1 | 62.3 | 62.8 | 62.2 | 62.2 | 63.8 | 62.7 | 63.5 | 62.1 | 62.6 | 63.5 | 100 | 62.8 | 62.5 | 63.3 | 61.5 | 61.8 | 61.4 | 62.2 | 60.9 | 60.9 | 61.1 | 60.8 | 60.7 | 60.6 | 60.2 |
| *Thalassotalea atypica* | 63.1 | 63.6 | 63.0 | 64.8 | 64.4 | 64.6 | 64.8 | 64.3 | 64.8 | 64.3 | 64.5 | 68.5 | 64.6 | 67.0 | 62.9 | 64.4 | 64.9 | 62.8 | 100 | 64.0 | 64.3 | 63.6 | 63.7 | 63.1 | 63.4 | 61.9 | 63.2 | 63.4 | 62.9 | 60.2 | 60.3 | 60.9 |
| *Litorilituus lipolyticus* | 63.1 | 63.3 | 62.7 | 64.2 | 64.4 | 64.6 | 65.2 | 64.6 | 65.0 | 64.5 | 65.4 | 64.0 | 65.0 | 62.9 | 63.4 | 75.6 | 63.2 | 62.5 | 64.0 | 100 | 62.5 | 70.3 | 63.9 | 70.1 | 64.0 | 68.3 | 70.4 | 70.3 | 69.8 | 59.5 | 59.6 | 60.2 |
| *Thalassotalea loyana* | 62.8 | 63.0 | 62.4 | 63.4 | 63.5 | 63.1 | 63.8 | 63.5 | 62.9 | 63.1 | 62.5 | 65.1 | 62.7 | 64.3 | 62.1 | 62.5 | 79.9 | 63.3 | 64.3 | 62.5 | 100 | 62.2 | 62.0 | 61.6 | 61.7 | 60.9 | 61.0 | 61.0 | 61.0 | 59.9 | 60.3 | 60.5 |
| *Colwellia ponticola* | 62.7 | 62.8 | 62.5 | 64.4 | 64.7 | 64.6 | 64.9 | 66.9 | 66.1 | 66.0 | 67.1 | 62.7 | 65.4 | 62.8 | 64.1 | 70.7 | 62.6 | 61.5 | 63.6 | 70.3 | 62.2 | 100 | 63.6 | 77.6 | 63.3 | 74.9 | 79.9 | 80.1 | 77.0 | 59.0 | 59.2 | 59.7 |
| *Thalassomonas actinarium* | 62.4 | 62.8 | 61.5 | 64.2 | 64.3 | 63.7 | 64.7 | 64.5 | 64.9 | 64.2 | 64.4 | 63.6 | 64.5 | 62.5 | 63.7 | 64.0 | 63.1 | 61.8 | 63.7 | 63.9 | 62.0 | 63.6 | 100 | 63.5 | 84.8 | 61.7 | 62.7 | 62.8 | 62.2 | 59.6 | 59.8 | 60.0 |
| *Colwellia marinimaniae* | 62.4 | 62.1 | 61.7 | 63.9 | 63.7 | 63.8 | 64.6 | 64.7 | 64.7 | 64.9 | 65.6 | 62.8 | 65.2 | 62.0 | 63.7 | 70.6 | 62.3 | 61.4 | 63.1 | 70.1 | 61.6 | 77.6 | 63.5 | 100 | 62.9 | 74.3 | 80.0 | 79.9 | 81.9 | 58.6 | 58.8 | 59.6 |
| *Thalassomonas viridans* | 62.3 | 62.7 | 61.4 | 64.1 | 63.9 | 63.2 | 64.2 | 63.8 | 64.6 | 63.7 | 64.0 | 63.3 | 64.3 | 62.4 | 62.5 | 63.9 | 62.4 | 62.2 | 63.4 | 64.0 | 61.7 | 63.3 | 84.8 | 62.9 | 100 | 61.4 | 62.2 | 62.4 | 62.1 | 59.1 | 59.9 | 59.9 |
| *Colwellia echini* | 61.7 | 61.7 | 60.6 | 62.8 | 63.6 | 63.5 | 63.2 | 64.3 | 64.2 | 64.5 | 64.9 | 61.8 | 63.4 | 61.3 | 63.6 | 67.8 | 61.5 | 60.9 | 61.9 | 68.3 | 60.9 | 74.9 | 61.7 | 74.3 | 61.4 | 100 | 75.4 | 75.5 | 74.2 | 58.1 | 58.2 | 59.1 |
| *Colwellia psycherythraea* | 61.3 | 61.8 | 61.0 | 63.3 | 63.8 | 63.6 | 64.4 | 65.2 | 65.0 | 64.6 | 66.7 | 62.2 | 65.3 | 61.5 | 63.2 | 70.5 | 61.7 | 60.9 | 63.2 | 70.4 | 61.0 | 79.9 | 62.7 | 80.0 | 62.2 | 75.4 | 100 | 92.4 | 79.0 | 58.4 | 58.1 | 59.7 |
| *Colwellia demingiae* | 61.3 | 62.0 | 61.1 | 63.4 | 63.9 | 63.4 | 64.2 | 65.1 | 65.5 | 64.5 | 66.6 | 62.4 | 65.5 | 61.7 | 63.2 | 70.5 | 61.9 | 61.1 | 63.4 | 70.3 | 61.0 | 80.1 | 62.8 | 79.9 | 62.4 | 75.5 | 92.4 | 100 | 79.5 | 58.7 | 58.3 | 59.8 |
| *Colwellia piezophila* | 61.2 | 61.5 | 60.8 | 63.0 | 63.6 | 63.1 | 64.2 | 64.6 | 64.2 | 64.1 | 65.4 | 62.1 | 64.0 | 61.2 | 62.6 | 70.1 | 61.6 | 60.8 | 62.9 | 69.8 | 61.0 | 77.0 | 62.2 | 81.9 | 62.1 | 74.2 | 79.0 | 79.5 | 100 | 58.3 | 58.2 | 60.1 |
| *Thalassotalea litorea* | 60.1 | 60.2 | 59.7 | 60.6 | 60.4 | 60.2 | 60.2 | 60.0 | 59.7 | 59.7 | 59.4 | 60.0 | 59.6 | 60.6 | 58.7 | 59.5 | 60.6 | 60.7 | 60.2 | 59.5 | 59.9 | 59.0 | 59.6 | 58.6 | 59.1 | 58.1 | 58.4 | 58.7 | 58.3 | 100 | 80.4 | 64.1 |
| *Thalassotalea mangrovi* | 59.8 | 60.2 | 59.7 | 60.8 | 60.2 | 60.1 | 60.3 | 59.8 | 59.4 | 59.7 | 59.3 | 60.7 | 59.6 | 60.5 | 58.8 | 59.7 | 60.4 | 60.6 | 60.3 | 59.6 | 60.3 | 59.2 | 59.8 | 58.8 | 59.9 | 58.2 | 58.1 | 58.3 | 58.2 | 80.4 | 100 | 64.6 |
| *Thalassotalea crassostreae* | 59.8 | 60.4 | 59.9 | 61.0 | 60.6 | 60.2 | 60.8 | 60.6 | 61.0 | 60.6 | 60.3 | 61.0 | 60.8 | 60.4 | 59.7 | 60.4 | 60.9 | 60.2 | 60.9 | 60.2 | 60.5 | 59.7 | 60.0 | 59.6 | 59.9 | 59.1 | 59.7 | 59.8 | 60.1 | 64.1 | 64.6 | 100 |

monophyletic clade, as at least five monophyletic clades were observed. Additionally, the genus *Colwellia* also did not form a monophyletic clade. Considering these results in combination with AAI values from the last section altogether implies that family *Colwelliaceae* might be a subject of reclassification in the future.

## Phenotypic characterization

PTE2$^T$ shared some biochemical features with *Thalassotalea* species, such as growth at 15, 25 and 30°C, the ability to hydrolyze Tween80, gelatin and DNA. Strain PTE2$^T$ was distinguished from other members with a total of 36 traits (growth at 4, 37 and 40°C, growth in 0% NaCl, oxidase, catalase, indole production, hydrolysis of starch, alginate and agar and 25 carbon assimilation tests) (Table 2).

All strains showed growth at 15°C, 25°C and 30°C and NaCl concentration of 1%, 3%, 6%, 8%, and 10%. All strains were able to hydrolyze Tween80, gelatin, and DNA. All strains tested positive for nitrate reduction. All strains tested negative for utilization of sucrose, melibiose, lactose, D-gluconate, N-acetylglucosamine, fumarate, citrate, aconitate, meso-erythritol, D-mannitol, glycerol, L-tyrosine, D-sorbitol, α-ketoglutarate, xylose, trehalose, glucuronate, δ-aminovalerate, cellobiose, putrescine, propionate, amygdalin, arabinose, D-galacturonate, glycerate, D-raffinose, L-rhamnose, D-ribose, salicine, DL-lactate, L-alanine and histidine.

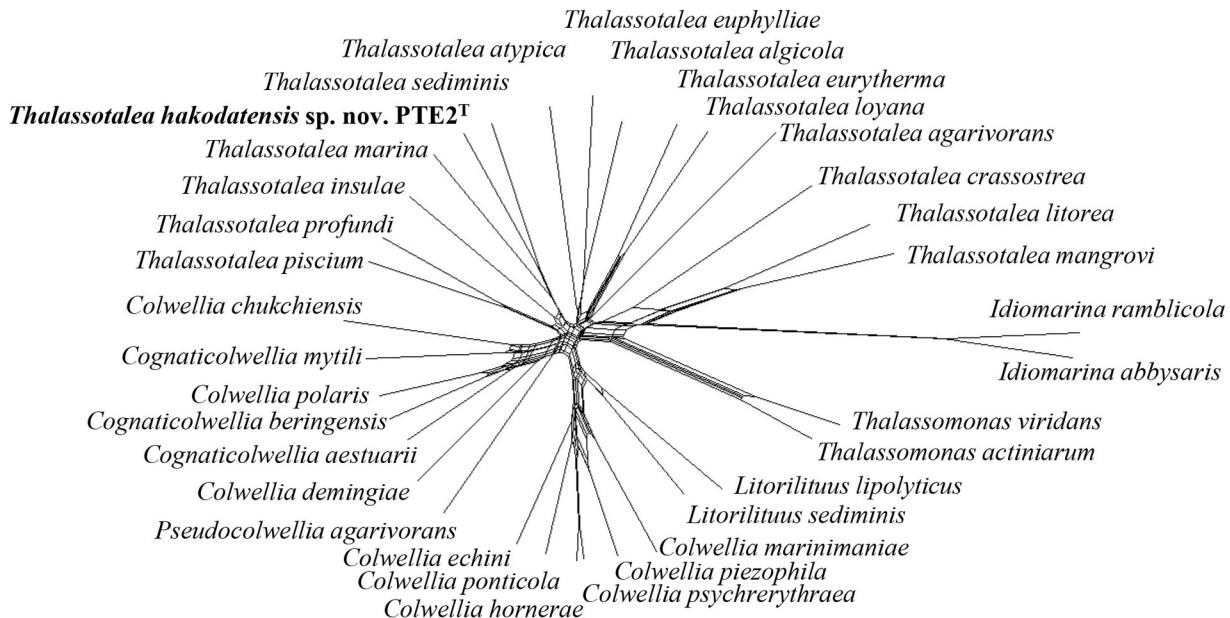

**Fig 3. MLSA network.** Bar, 0.01 substitutions per nucleotide position. Topology of the clade which PTE2 was belonged to was supported by a 100% bootstrap value.

## Pangenomic analysis

The pangenome of *Thalassotalea* species consists of 17,797 gene clusters (53,489 genes) (Fig 4). Genes were classified into *Core* for the genes present in all strains, and species-specific bins for the genes that are unique for each species. *Core* consisted of 1,175 gene clusters (17,979 genes).

N-acyl homoserine lactone hydrolase gene was present in PTE2[T], *T. sediminis* and *T. insulae*. The enzyme shows the quorum-quenching activity of hydrolyzing N-acyl homoserine lactones (AHLs), a signal molecule used by many Gram-negative bacteria in quorum-sensing pathways [51]. This activity of quorum quenching has been shown to strengthen host animal resistance to pathogenic bacteria by disrupting their quorum-sensing activities [52].

Strain PTE2[T] had two genes for with alginate degradation enzyme, namely, poly-(β-D-mannuronate) lyase and oligo-alginate lyase. The phenotype test also show that this strain has alginate hydrolysis ability (Table 2). The presence of these genes implies that strain PTE2[T] may aid digestion of alginate in sea cucumber gut. In previous study, it was observed that bacteria associated with the algal polysaccharide degradation were abundant in fast-growing *Apostichopus japonicus* compared to slow-growing individuals, and such bacteria are believed to assist host in obtaining energy [53].

## *In silico* chemical taxonomy

The cellular fatty acid (FA) profile of *Thalassotalea* species is reported to consist mainly of even-numbered linear mono-unsaturated or saturated chains (i.e. C14:0, C16:0, C16:1 and C18:1), with small amount of odd-numbered linear mono-unsaturated or saturated chains (i.e. C13:0, C15:1 and C17:0), 3-hydroxy FAs (i.e. C11:0 3-OH, C12:0 3-OH) and branched-chain FAs (i.e. iso-C14:0, iso-C16:0) (S3 Table) [1, 3, 4, 8, 11, 13, 15–18]. Pangenomic analysis among described *Thalassotalea* species reconstructed the basic type II fatty acid biosynthesis

**Table 2. Phenotypic characteristics of PTE2[T] and *Thalassotalea* reference strains.**

| Characteristics | 1 | 2 | 3 | 4 | 5 | 6 | 7 | 8 |
|---|---|---|---|---|---|---|---|---|
| Growth at | | | | | | | | |
| 4°C | + | - | + | + | - | + | - | - |
| 37°C | - | + | - | + | + | - | + | + |
| 40°C | - | NT | - | - | + | - | - | NT |
| Growth in NaCl (broth) | | | | | | | | |
| 0% | - | NT | + | - | + | - | + | NT |
| OF test | O | O | N | O | O | O | O | O |
| Oxidase | + | - | + | + | + | + | + | + |
| Catalase | + | + | + | w | + | w | w | + |
| Indole production | - | NT | - | - | - | w | - | NT |
| Hydrolysis of | | | | | | | | |
| Starch | + | - | - | + | + | + | + | - |
| Alginate | + | - | - | - | w | - | - | - |
| Agar | - | - | - | - | + | - | - | - |
| Antibiotic susceptibility | | | | | | | | |
| GM120(Gentamicin 120) | + | NT | +[*1] | + | +[*1] | NT | + | NT |
| GAT5(Gemifloxacin 5) | + | NT | NT | + | NT | NT | + | NT |
| GTX30(Cefotaxime 30) | + | NT | NT | + | NT | NT | + | NT |
| CB100(Carbenicillin 100) | + | NT | NT | + | +[*1] | +[*3] | + | NT |
| CLR15(Clarithromycin 15) | + | NT | -[*2] | + | NT | -[*3] | + | NT |
| SXT(Sulfamethoxazole/Trimethoprim) | + | NT | NT | + | NT | NT | + | NT |
| AM10(Ampicillin 10) | w | NT | +[*1] | + | +[*1] | +[*3] | + | NT |
| Utilization of | | | | | | | | |
| D-Mannose | w | - | - | - | - | - | - | - |
| D-Galactose | + | + | - | - | - | - | + | - |
| D-Fructose | + | - | - | - | - | - | - | - |
| Maltose | - | + | - | - | - | - | - | - |
| N-Acetylglucosamine | - | + | - | - | - | - | + | - |
| Succinate | + | - | - | - | - | - | + | - |
| γ-Aminobutyrate | - | - | - | - | - | + | - | - |
| Xylose | + | - | - | - | - | - | - | - |
| D-Glucose | + | + | - | - | - | - | w | - |
| Acetate | - | + | - | - | - | - | + | - |
| D-Glucosamine | - | + | w | - | - | - | + | - |
| Pyruvate | + | + | - | - | - | - | - | - |
| Cellobiose | - | + | - | - | - | - | - | - |
| L-Proline | + | + | - | - | - | + | + | - |
| L-Glutamate | + | + | - | - | - | + | + | - |
| Putrescine | - | - | - | - | - | + | - | - |
| Salicine | - | + | - | - | - | - | - | - |
| DL-Lactate | - | + | - | - | - | - | + | - |
| L-Arginine | + | + | - | - | - | + | + | - |
| L-Asparagine | + | - | - | - | - | + | + | - |
| L-Citrulline | - | - | - | - | - | + | - | - |
| Glycine | + | - | - | - | - | - | - | - |
| L-Histidine | - | + | - | - | - | - | - | - |
| L-Ornithine | + | - | - | - | - | - | - | - |

*(Continued)*

**Table 2.** (Continued)

| Characteristics | 1 | 2 | 3 | 4 | 5 | 6 | 7 | 8 |
|---|---|---|---|---|---|---|---|---|
| L-Serine | + | - | - | - | - | - | - | - |

Strains: 1, *T. hakodatensis* sp. nov. PTE2[T], 2, *T. sediminis* KCTC 42588 [T], 3, *T. piscium* JCM 18590 [T], 4, *T. marina* KCTC 42731[T], 5, *T. agarivorans* JCM 13379 [T], 6, *T. atypica* JCM 31894 [T], 7, *T. loyana* LMG 22536 [T], 8, *T. eurytherma* JCM 18482.

[*1]: data from [6],

[*2]: data from [13],

[*3]: data from [15].

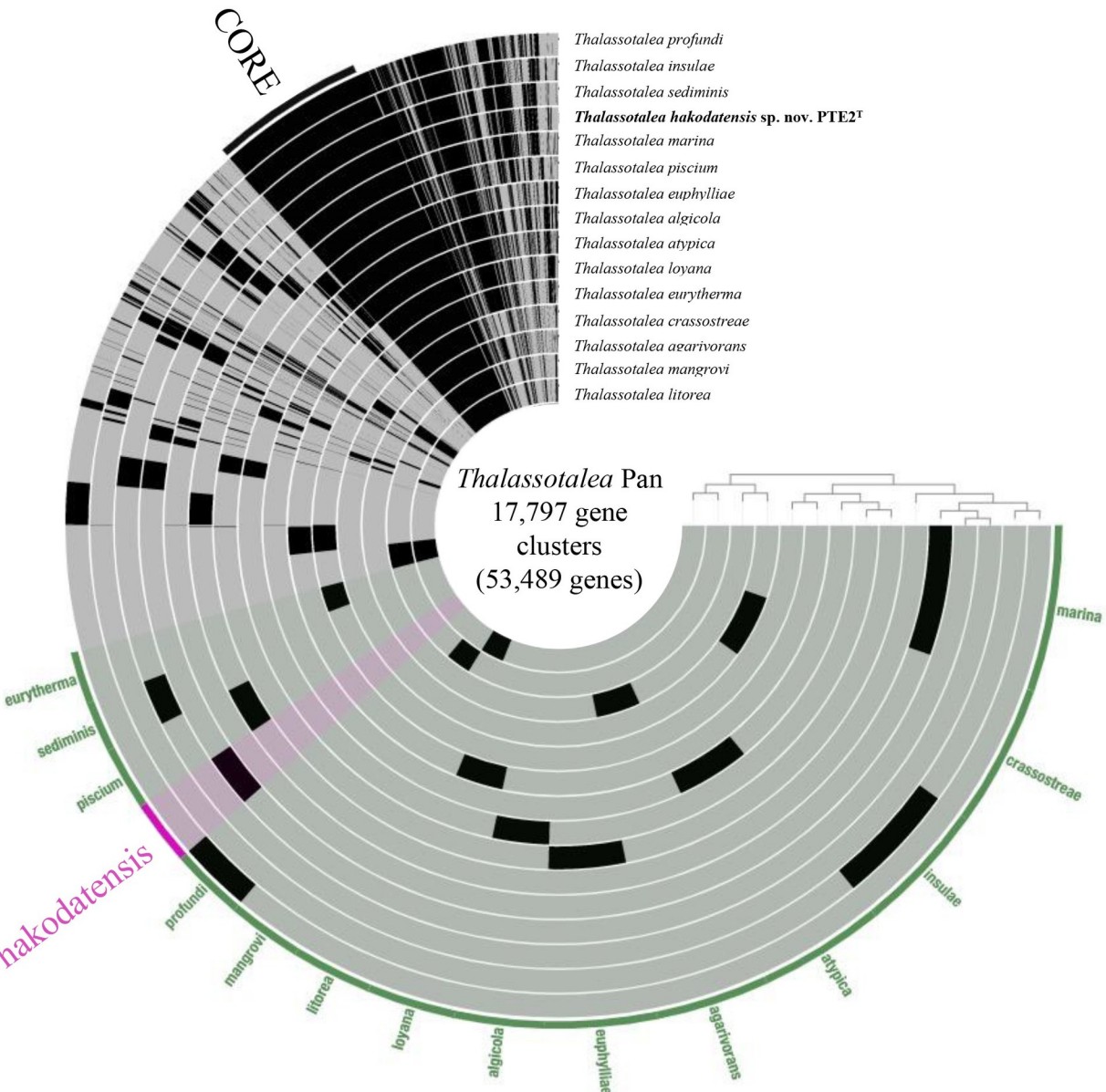

**Fig 4. Anvi'o representation of the pangenome of the *Thalassotalea* species.** Layers represent each genome, and the bars represent the occurrence of gene clusters. The darker colored areas of the bars belong to *Core* or PTE2[T] specific bin.

(FAS II) pathway driven by FabABFDGVYZ and AccABCD, which is very similar to that of *E. coli* [54] (S4 Table and S1 Fig). The FAS II pathway could contribute to even-numbered linear mono-unsaturated or saturated chains (C14:0, C16:0, C16:1 ω7c of summed features 3 and C18:1 ω7c), which occupied approximately 20–60% of the total fatty acid (S3 Table). C16:0 is one of the major products produced from the FAS II pathway, which means PTE2[T] could produce C16:0 (S1 Fig). ω7c mono-unsaturated fatty acids (C16:1 ω7c and C18:1 ω7c) are also major in this of bacteria and can be produced through ω7 mono-unsaturated fatty acid synthesis initiated by isomerization of trans-2-decenoyl-ACP into cis-3-decenoyl-ACP by FabA (S1 Fig). After elongation by FabB, the acyl chain is returned to the FASII pathway and goes through further elongation, producing C16:1ω7c and C18:1ω7c [55]. All strains, including PTE2[T] have *fabA* and *fabB*, thus it is suggesting that PTE2[T] is also capable of producing C16:1ω7c and C18:1ω7c.

Odd-numbered saturated fatty acids can also be produced by the FAS II pathway by using propanoyl-CoA as a primer molecule for the initial condensation, instead of acetyl CoA used for even-numbered saturated fatty acids [56]. Amongst multiple pathways for the synthesis of propanoyl-CoA, pangenome analysis in this study revealed that all 15 strains possess *bkdAAB* genes which are responsible for producing propanoyl-CoA from 2-oxobutanoate via 1-hydroxy propyl-Thpp and S-propanyl-dihydrolipoamide-E. Moreover, 10 strains including PTE2[T] had genes responsible to another propanoyl-CoA production pathway, *accABCD*, *fadB*, *fadJ* and *acuI*, which uses acetyl-CoA (S4 Table). These results provide evidence that strain PTE2[T] has the capacity to synthesize propanoyl-CoA, a precursor for the synthesis of odd-numbered saturated fatty acids. C13:0 and C17:0 derived from this pathway consist approximately 0–18% of the *Thalassotalea* fatty acid profile (S3 Table). Notably, C17:0 plays a vital role in the production of heptadecenoyl-CoA (C17:1 ω8c), which is described in further detail below.

3-hydroxylated FAs such as C11:0 3-OH and C12:0 3-OH, which are the primary fatty acids in lipid A as well as in ornithine-containing lipids, could be supplied by the FAS II pathway, since 3-hydroxy-acyl-ACP is known to be normally intermediated in the FAS II elongation cycle [55]. *Thalassotalea* core genes also included *lpxA* and *lpxD*, a gene responsible for the incorporation of 3-hydroxy-acyl chains to UDP-N-acetyl-alpha-D-glucosamine and UDP-3-O-(3-hydroxytetradecanoyl)-D-glucosamine, respectively, which are essential for the biosynthesis of lipid A [57, 58].

Fatty acid desaturase (Des) catalyzes to produce unsaturated fatty acid by introducing double bonds to the fatty acid under aerobic conditions [59]. Fatty acid desaturase, *des1* was found in the core gene. 3D-structure prediction using Phyre2 program shows that these enzymes are likely to be stearoyl-CoA desaturase (SCD) (S5 Table). SCD introduces *cis* double bond at the Δ9 position of palmitoyl-CoA (C16:0), heptadecanoyl-CoA (C17:0) or stearoyl-CoA (C18:0), producing palmitoleoyl-CoA (C16:1 ω7c), heptadecenoyl-CoA (C17:1 ω8c) or oleoyl-CoA (C18:1 ω9c) [60]. Using this enzyme, *Thalassotalea* species could produce C17:1 ω8c, a major fatty acid consisting approximately 10–25% of the total fatty acid profile (S3 Table).

The biosynthesis pathway responsible for producing linear fatty acids also produces branched-chain fatty acids, but with distinct primer molecules. Iso-fatty acids with even-numbered chains found in *Thalassotalea* species are produced using 3-methylbutyryl-CoA as a primer molecule instead of acetyl-CoA, which is used for production of linear fatty acids. All strains in this study contain the *ilvE* and *dkdAAB* genes that are involved in the degradation of L-leucine to 3-methylbutyryl-CoA, implicating that these species have the potential to produce even-numbered iso-fatty acids such as iso-C14:0 and iso-C16:0 (S1 Fig).

The genome of strain PTE2[T] was subjected to a comparative analysis to investigate the presence of genes involved in the fatty acid synthesis type II (FAS II) pathway, showing stain

PTE2$^T$ shared a core gene set that was highly similar to that of other 14 strains (S1 Fig and S4 Table). Genomic structures of FAS II core genes are likely to be retained among described *Thalassotalea* species (S2 and S3 Figs), which could lead to the conclusion that the novel strain is capable of producing similar FA profiles as *Thalassotalea* reference strains, consisting of mainly even-numbered linear mono-unsaturated or saturated chains (i.e. C16:0, C16:1 and C18:1), odd-numbered linear mono-unsaturated or saturated chains (i.e. C15:1 and C17:0), 3-hydroxy FAs (i.e. C11:0 3-OH, C12:0 3-OH) and branched-chain FAs (i.e. iso-C14:0, iso-C16:0). However, a pathway to produce some of the major fatty acids such as C15:1 ω8c and iso-C17:1 ω5C could not be identified.

Pangenomic analysis among 15 strains also revealed a comprehensive gene set for the biosynthesis of phosphatidylglycerol (PG) and phosphatidylethanolamine (PE); *plsX*, *plsY*, *plsC*, *cdsA*, *pssA*, *psd*, *pgsA* and *pgpA*, for all the strains (S6 Table). This result indicates that PTE2$^T$ produces PG and PE as major polar lipids, similar to other 14 reference species. Interestingly, *clsA/B* and *clsC* genes, which are responsible for the production of diphosphatidylglycerol (DPG) [61], were detected in the genomes of four strains, *T. piscium*, *T. agarivorans*, *T. crossostreae* and *T. litorea*. This result suggests that these four strains are potential DPG producers.

The only respiratory quinone reported from previously described *Thalassotalea* species is ubiquinone-8 (Q-8) (S3 Table). Core genes of 15 strains include *ubi* genes responsible for the biosynthetic pathway (*ubiC*, *ubiA*, *ubiD*, *ubiX*, *ubiI*, *ubiG*, *ubiH*, *ubiE*), and *ispAB*. In addition to those, core genes also include three genes (*ubiB*, *ubiJ*, *ubiK*) coding accessory proteins also required for Q-8 biosynthesis, but with rather hypothetical functions. These results suggest that the predominant ubiquinone of PTE2$^T$ is Q-8.

## Conclusions

Using the results of modern genome taxonomic studies combined with classical phenotyping, which fulfills phylogenetic, genomic, and phenotypic cohesions, we propose the strain PTE2$^T$ as *Thalassotalea hakodatensis* sp. nov. (PTE2$^T$ = JCM 34608$^T$ = KCTC 82592$^T$), a novel species in the genus *Thalassotalea*.

### Description of *Thalassotalea hakodatensis* sp. nov.

*Thalassotalea hakodatensis* sp. nov. (ha.ko.da.ten'sis. N.L. fem. adj. *hakodatensis*, from Hakodate, referring to the isolation site of the strain).

Gram-negative, rod-shaped and motile. Colonies on MA are yellowish-white, 0.5–0.75 mm in diameter after culture for 3 days. No pigmentation and bioluminescence are observed. The DNA G+C content is 38.4%, and genome size is 4.28 Mb. Growth occurs at 4˚C, 15˚C, 25˚C and 30˚C, with NaCl concentrations of 1%, 3%, 6%, 8% and 10%. OF-test oxidative. susceptible for ampicillin (10 µg), cefotaxime (30 µg), gatifloxacin (5 µg), carbenicillin (100 µg), clarithromycin (15 µg) and sulfamethoxazole/trimethoprim. Positive for oxidase- and catalase- test, nitrate reduction, hydrolysis of starch, alginate, tween 80, gelatin and DNA, utilization of D-mannose, D-galactose, D-fructose, succinate, xylose, D-glucose, pyruvate, L-proline, L-glutamate, L-arginine, L-asparagine, glycine, L-ornithine and L-serine. Negative for production of indole, hydrolysis of agar, utilization of sucrose, maltose, melibiose, lactose, D-gluconate, N-acetylglucosamine, fumarate, citrate, aconitate, meso-erythritol, D-mannitol, glycerol, γ-aminobutyrate, L-tyrosine, D-sorbitol, DL-malate, α-ketoglutarate, trehalose, glucuronate, acetate, D-glucosamine, δ-aminovalerate, cellobiose, putrescine, propionate and amygdalin.

The type strain PTE2$^T$ (= JCM 34608$^T$ = KCTC 82592$^T$) was isolated from a pentactula larvae of *Apostichopus japonicus* reared in a laboratory aquarium at Hokkaido University, Hokkaido, Japan. The GenBank accession number for the 16S rRNA gene sequence of the type

strain is LC757706. The complete genome sequence of the strain is deposited in the DDBJ/ENA/GenBank under the accession number AP027365.

## Supporting information

**S1 Table. List of other *Colwelliaceae* genomes used for genome taxonomy of PTE2[T].** (PDF)

**S2 Table. *In silico* DDH and ANI values of *Thalassotalea hakodatensis* PTE2[T] sp. nov. against *Thalassotalea* species.** (PDF)

**S3 Table. Fatty acid, isoprenoid quinone and polar lipid profile of previously reported *Thalassotalea*.** Strains: 1, *T. sediminis* KCTC 42588 [T], 2, *T. insulae* KCTC 42588 [T], 3, *T. piscium* JCM 18590 [T], 4, *T. marina* KCTC 42731[T], 5, *T. profundi* YM155[T], 6, *T. agarivorans* JCM 13379 [T], 7, *T. eurytherma* JCM 18482[T], 8, *T. atypica* JCM 31894 [T], 9, *T. mangrovi* zs-4[T], 10, *T. crassostreae* LPB 0090[T],11, *T. loyana* LMG 22536[T], 12, *T. algicola* M1531[T], 13, *T. litorea* HMF4135[T], 14, *T. euphylliae* Eup-16[T]. (PDF)

**S4 Table. FAS associated genes composition of 15 species.** (PDF)

**S5 Table. Results of 3D-structure prediction of Des1 by Phyre2.** (PDF)

**S6 Table. PG, PE and DPG associated genes composition of each strain.** (PDF)

**S1 Fig. Predicted fatty acid synthetic pathway in *Thalassotalea* species.** ACP: acyl-carrier protein; AccABCD: acetyl-CoA carboxylase complex; FabD: malonyl-CoA: ACP transacylase; FabH/FabY: 3-ketoacyl-ACP synthase III; FabB: 3-ketoacyl-ACP synthase I; FabF: 3-ketoacyl-ACP synthase II; FabG: 3-ketoacyl-ACP reductase; FabA: 3-hydroxyacyl-ACP dehydratase/trans-2-decenoyl-ACP isomerase; FabV enoyl-ACP reductase; FabZ: 3-hydroxyacyl-ACP dehydratase. (PDF)

**S2 Fig. Genomic structure of *Thalassotalea fab* and associated genes.** (PDF)

**S3 Fig. Genomic distribution of *fab* and associated genes (only strains with complete genome sequence).** Protein/enzyme name each gene is coding: *fabA*: 3-hydroxyacyl-ACP dehydrase/trans-2-decenoyl-ACP isomerase; *fabD*: malonyl-CoA: ACP transacylase; *fabF*: 3-ketoacyl-ACP synthase ; *fabG*: 3-ketoacyl-ACP reductase; *fabH*: 3-ketoacyl-ACP synthase ; *fabV* enoyl-ACP reductase; *fabY*: 3-ketoacyl-ACP synthase; *fabZ*: 3-hydroxyacyl-ACP dehydratase; *accABCD*: carboxylase complex; *plsX*: phosphate acyltransferase; *acpP*: Acyl-carrier protein. (PDF)

## Acknowledgments

We gratefully thank for Professor (Emeritus) Aharon Oren, The Hebrew University of Jerusalem, for his advice on bacterial names.

## Author Contributions

**Conceptualization:** Sayaka Mino, Tomoo Sawabe.

**Data curation:** Ryota Yamano, Juanwen Yu, Chunqi Jiang, Sayaka Mino, Tomoo Sawabe.

**Formal analysis:** Ryota Yamano, Juanwen Yu, Chunqi Jiang, Tomoo Sawabe.

**Funding acquisition:** Tomoo Sawabe.

**Investigation:** Ryota Yamano, Juanwen Yu, Alfabetian Harjuno Condro Haditomo, Chunqi Jiang, Tomoo Sawabe.

**Methodology:** Ryota Yamano, Juanwen Yu, Chunqi Jiang, Sayaka Mino, Yuichi Sakai, Tomoo Sawabe.

**Project administration:** Tomoo Sawabe.

**Resources:** Juanwen Yu, Sayaka Mino, Jesús L. Romalde, Kyuhee Kang, Tomoo Sawabe.

**Supervision:** Sayaka Mino, Yuichi Sakai, Tomoo Sawabe.

**Writing – original draft:** Ryota Yamano, Juanwen Yu, Sayaka Mino, Tomoo Sawabe.

**Writing – review & editing:** Ryota Yamano, Juanwen Yu, Alfabetian Harjuno Condro Haditomo, Chunqi Jiang, Sayaka Mino, Jesús L. Romalde, Kyuhee Kang, Yuichi Sakai, Tomoo Sawabe.

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
