## [Decision Letter · Decision Letter 0]

18 Apr 2023

PONE-D-23-07380Genome taxonomy of the genus Thalassotalea and proposal of Thalassotalea hakodatensis sp. nov. isolated from sea cucumber larvaePLOS ONE

Dear Dr. Sawabe,

Thank you for submitting your manuscript to PLOS ONE. After careful consideration, we feel that it has merit but does not fully meet PLOS ONE’s publication criteria as it currently stands. Therefore, we invite you to submit a revised version of the manuscript that addresses the points raised during the review process.

Reviewers have now commented on your paper. You will see that one of the reviewers are advising that you revise your manuscript. When revising your work, please submit a list of changes or a rebuttal against each point which is being raised when you submit the revised manuscript.The reviewers' comments can be found at the end of this email.

We look forward to receiving your revised manuscript.

Kind regards,

Cristiane Thompson

Academic Editor

PLOS ONE

Journal Requirements:

https://researchers.general.hokudai.ac.jp/profile/en.24b04e78bac629fd520e17560c007669.html?mode=pc

https://journals.plos.org/plosone/article?id=10.1371%2Fjournal.pone.0271174

In your revision ensure you cite all your sources (including your own works), and quote or rephrase any duplicated text outside the methods section. Further consideration is dependent on these concerns being addressed

Reviewers' comments:

Reviewer's Responses to Questions

**Comments to the Author**

1. Is the manuscript technically sound, and do the data support the conclusions?

Reviewer #1: Yes

Reviewer #2: Yes

2. Has the statistical analysis been performed appropriately and rigorously? 

Reviewer #1: Yes

Reviewer #2: N/A

3. Have the authors made all data underlying the findings in their manuscript fully available?

Reviewer #1: Yes

Reviewer #2: Yes

4. Is the manuscript presented in an intelligible fashion and written in standard English?

Reviewer #1: Yes

Reviewer #2: Yes

5. Review Comments to the Author

Reviewer #1: The authors present a sound body of evidence to justify the description of this new species of the genus Thalassotalea.

It is a good example of genomic taxonomy. I am convinced that this format is adequate for the description of a new species.

Reviewer #2: The study was well conducted, and the results are very interesting. Congratulations.However, I have some points that need to be addressed.

First, please check the figure´s quality. "suggesting strain PTE2 may pale an important role in host-microbe interaction during sea cucumber development"Did the authors mean "may play"?

The 16S sequence was generated using which technology? How these sequences were processed? Please clarify.It is not clear what genomes were used in OrthoANI and DDH analysis. What were the parameters used in MLSA phylogeny? ML? NJ? Bootstrap?Do you know if the Figure 1 subtitle is correct? Because Idiomarina rambicola is not visible in the figure. 

How did the MLSA network build? I could not find the section in Methodology.I suggest that the phrase below could be written removing "I was not able"."However, I was not able to identify a pathway to produce some of the major fatty acids such as C15:1 ω8c and iso-C17:1 ω5C."

6. PLOS authors have the option to publish the peer review history of their article (what does this mean?). If published, this will include your full peer review and any attached files.

Reviewer #1: No

Reviewer #2: No

---

## [Author Response · Author response to Decision Letter 0]

11 May 2023

Dear Professor Thompson, the editor, and reviewers,

We appreciate editor and reviewers for constructive suggestions for PONE-D-23-07380. We improved the manuscript according to the editorial office and reviewers’ comments. Responses for specific comments are described as follows. All changes were highlighted in yellow. 

Editorial office

1. Please ensure that your manuscript meets PLOS ONE's style requirements, including those for file naming. The PLOS ONE style templates.

Response: Thank you so much for careful check. Style was fit to PLOS ONE format.

2. We noticed you have some minor occurrence of overlapping text with the following previous publication(s), which needs to be addressed.

Response: Thank you for the suggestion. We used almost same approach, so many parts of body text were overlapped. We did remove redundancy and put the previous paper as reference.

Reviewer: 1

1. The authors present a sound body of evidence to justify the description of this new species of the genus Thalassotalea. It is a good example of genomic taxonomy. I am convinced that this format is adequate for the description of a new species.

Response: Thank you so much for the positive comments.

Reviewer: 2

1. The study was well conducted, and the results are very interesting. Congratulations. However, I have some points that need to be addressed. First, please check the figure´s quality.

Response: Thank you for the comments. We tried to increase figs resolution.

2. "suggesting strain PTE2 may pale an important role in host-microbe interaction during sea cucumber development" Did the authors mean "may play"?

Response: Thank you so much for the suggestion. We did correct it.

3. The 16S sequence was generated using which technology? How these sequences were processed? Please clarify.

Response: Thank you for the comments. We put the methodology.

4. It is not clear what genomes were used in OrthoANI and DDH analysis.

Response: Thank you for the comments. We put the option description; calculate OrthoANI with a default setting and DDH by formula 2.

5. Do you know if the Figure 1 subtitle is correct? Because Idiomarina rambicola is not visible in the figure.

Response: Idiomarina was used for outgroup, but after figuring rooted subtree shown in Fig 1, Idiomarina position was not visible. We added some explanation in Fig 1 title.

6. What were the parameters used in MLSA phylogeny? ML? NJ? Bootstrap? 

How did the MLSA network build? I could not find the section in Methodology.

Response: Thank you for the comments. We also put some Idiomarina sequences in Fig 3, and options for MLSA.

6. I suggest that the phrase below could be written removing "I was not able". "However, I was not able to identify a pathway to produce some of the major fatty acids such as C15:1 ω8c and iso-C17:1 ω5C." 

Response: Thank you for the comments. According to the suggestion, we did correct the sentence.

---

## [Editor Report · Decision Letter 1]

22 May 2023

Genome taxonomy of the genus Thalassotalea and proposal of Thalassotalea hakodatensis sp. nov. isolated from sea cucumber larvae

PONE-D-23-07380R1

Dear Dr. Sawabe,

We’re pleased to inform you that your manuscript has been judged scientifically suitable for publication and will be formally accepted for publication once it meets all outstanding technical requirements.

Kind regards,

Cristiane Thompson

Academic Editor

PLOS ONE

---

## [Editor Report · Acceptance letter]

24 May 2023

PONE-D-23-07380R1 

Genome taxonomy of the genus *Thalassotalea* and proposal of *Thalassotalea hakodatensis* sp. nov. isolated from sea cucumber larvae 

Dear Dr. Sawabe:

I'm pleased to inform you that your manuscript has been deemed suitable for publication in PLOS ONE. Congratulations! Your manuscript is now with our production department. 

Kind regards, 

on behalf of

prof. Cristiane Thompson 

Academic Editor

PLOS ONE